🔓 | **Open Peer Review** | Antimicrobial Chemotherapy | Research Article

# Drug-induced stress mediates *Plasmodium falciparum* ring-stage growth arrest and reduces *in vitro* parasite susceptibility to artemisinin

**Lucien Platon,**[1,2,3,4] **Didier Leroy,**[5] **David A. Fidock,**[6,7] **Didier Ménard**[1,3,4,8]

**ABSTRACT** During blood-stage infection, *Plasmodium falciparum* parasites are constantly exposed to a range of extracellular stimuli, including host molecules and drugs such as artemisinin derivatives, the mainstay of artemisinin-based combination therapies currently used as first-line treatment worldwide. Partial resistance of *P. falciparum* to artemisinin has been associated with mutations in the propeller domain of the *Pfkelch13* gene, resulting in a fraction of ring stages that are able to survive exposure to artemisinin through a temporary growth arrest. Here, we investigated whether the growth arrest in ring-stage parasites reflects a general response to stress. We mimicked a stressful environment *in vitro* by exposing parasites to chloroquine or dihydroartemisinin (DHA). We observed that early ring-stage parasites pre-exposed to a stressed culture supernatant exhibited a temporary growth arrest and a reduced susceptibility to DHA, as assessed by the ring-stage survival assay, irrespective of their *Pfkelch13* genotype. These data suggest that temporary growth arrest of early ring stages may be a constitutive, *Pfkelch13*-independent survival mechanism in *P. falciparum*.

**IMPORTANCE** *Plasmodium falciparum* ring stages have the ability to sense the extracellular environment, regulate their growth, and enter a temporary growth arrest state in response to adverse conditions such as drug exposure. This temporary growth arrest results in reduced susceptibility to artemisinin *in vitro*. The signal responsible for this process is thought to be small molecules (less than 3 kDa) released by stressed mature-stage parasites. These data suggest that Pfkelch13-dependent artemisinin resistance and the growth arrest phenotype are two complementary but unrelated mechanisms of ring-stage survival in *P. falciparum*. This finding provides new insights into the field of *P. falciparum* antimalarial drug resistance by highlighting the extracellular compartment and cellular communication as an understudied mechanism.

**KEYWORDS** malaria, *P. falciparum*, ring stages, temporary cell growth arrest, artemisinin resistance, intercellular communication

Address correspondence to Lucien Platon, lplaton@unistra.fr, or Didier Ménard, dmenard@pasteur.fr.

The authors declare no conflict of interest.

See the funding table on p. 12.

Despite recent efforts and progress, *Plasmodium falciparum* malaria continues to have a significant impact on human health. In 2022, the number of *P. falciparum* infections was estimated to be 249 million, resulting in 608,000 deaths, mainly in sub-Saharan Africa (1). Today, artemisinin-based combination therapies (ACTs) are recommended as first-line treatment for uncomplicated *P. falciparum* malaria worldwide (2). Introduced in the 2000s, ACTs have had a major impact on reducing mortality (1). ACTs are a combination of fast-acting artemisinin derivatives (ARTs) that can reduce the biomass of drug-sensitive parasites by up to 10,000-fold within 48 h and slower-acting partner drugs such as lumefantrine, amodiaquine, mefloquine, or piperaquine that reduce the selective pressure for ART resistance and clear residual parasitemia (3, 4).

Unfortunately, the emergence and spread of artemisinin resistance in *P. falciparum* and the declining efficacy of ACTs, first reported in the Greater Mekong sub-region in 2009 and more recently in Papua New Guinea, Guyana, and sub-Saharan Africa (Rwanda, Uganda, Ethiopia, and Eritrea), are a major concern (5–21). A better understanding of the mechanisms of action and resistance to ACTs is urgently needed, not only to manage current clinical use but also to rationally design new strategies. ART-R is defined as a delayed clearance of circulating asexual blood-stage parasites following 3 days of ACT treatment or artemisinin monotherapy (half-life >5.5 h) (1). *In vitro*, ART-R manifests as increased survival of early ring blood-stage parasites (0–3 h post-invasion [hpi]) exposed to a pulse of 700 nM dihydroartemisinin (DHA, the active metabolite of all clinically used ARTs) for 6 h as measured by the ring-stage survival assay (RSA) (22). Subsequently, mutations in the beta-propeller domain of the *P. falciparum kelch* gene (*Pfkelch13*) were shown to confer ART-R *in vivo* and *in vitro* (5, 16). Currently, the most common *Pfkelch13* mutation associated with delayed clearance in the eastern part of the Greater Mekong sub-region is C580Y (9–11, 13, 14). Recent reports suggest that *Pfkelch13* mutations result in lower levels of the kelch13 protein, which is involved in the endocytosis and digestion of hemoglobin in the early ring-stage parasite (23–27). Therefore, it has been postulated that the mutations result in reduced endocytosis and degradation of hemoglobin in early ring-stage parasites, resulting in reduced levels of $Fe^{2+}$-heme available to activate ART (23, 24, 26). In addition, like other organisms, *P. falciparum* can mount protective responses to sudden changes in its environment such as temporary cell cycle arrest of a fraction of the ring-stage population (also called quiescence or dormancy) to increase survival under unfavorable conditions (28, 29). This has been demonstrated for some metabolic conditions (variations in sugars, amino acids, lipids, and micronutrients) and exposure to febrile temperatures (30–32) and also for drugs such as ART that have been shown to induce this phenotype *in vitro* (29, 33) and *in vivo* (34).

Here, we investigated whether the early ring parasites can sense a stressful environment and induce a fraction of the parasites to undergo a temporary growth arrest that can survive DHA exposure. Using the NF54$^{WT}$, 3D7$^{WT}$, 3D7$^{C580Y}$, and Cambodian *P. falciparum kelch13* wild-type (WT) and mutant (C580Y) parasite lines, we show that the culture medium prepared from chloroquine (CQ)-killed NF54$^{WT}$ parasites (mimicking a stressful environment) can induce a temporary growth arrest of *P. falciparum* early ring-stage parasites. The temporary growth arrest was dependent on the genetic background of the parasite lines but independent of the *Pfkech13* genotype. We confirmed that the proportion of ring stages capable of entering a growth arrest increased, resulting in an increased proportion of parasites that were able to survive a 700 nM pulse of DHA for 6 h. Reduced susceptibility to DHA was shown to be threshold-dependent, with the proportion of viable parasites decreasing significantly as the concentration of the stressful environment decreased. We also provide evidence that the signal is potentially mediated by several molecules of small molecular weights (<3 kDa), which would rule out a role for extracellular vesicles (EVs). Finally, we show that DHA-exposed susceptible mature stages of *P. falciparum* may release these molecules as they die. These potential signaling molecules could, thus, act as danger signals of environmental stress and influence ring-stage growth to optimize population survival, irrespective of the *Pfkelch13* genotype. This suggests that *Pfkelch13* mutations that confer artemisinin partial resistance and the ring-stage stress-sensing pathway may be two complementary survival mechanisms.

## RESULTS

### Stress-induced medium mediates delayed growth of *P. falciparum* ring stages *in vitro*

To investigate whether *P. falciparum* ring-stage parasites can sense their environment and respond to stress-induced conditions, we tested whether a culture medium obtained from a chloroquine-sensitive (CQ-S) parasite line (NF54 strain) exposed to 200 nM CQ for 20 h would affect the intraerythrocytic developmental cycle progression of the

chloroquine-resistant (CQ-R) *P. falciparum* parasite lines (Table S1). We produced the stress-induced medium by generating a culture medium (named M_NF54_CQ_20h) from an asynchronous NF54 CQ-S parasite line exposed to 200 nM CQ for 20 h. The killing effect of CQ on NF54 CQ-S blood stages was confirmed by the microscopic examination of a Giemsa-stained blood smear (Fig. S1 and S2). We then exposed 0–3 hpi ring stages of three Cambodian CQ-R *P. falciparum* parasite lines at 0.5% parasitemia to M_NF54_CQ20h or RPMI complete medium (as control) for 24 h (H24 time point). After two washes with incomplete RPMI medium, the parasites were cultured in a complete RPMI medium for a further 24 h (H48 time point). At H24 and H48, red blood cells(RBCs) were collected and used to prepare Giemsa-stained blood smears.

We observed that ring-stage parasites exposed to complete RPMI medium (control) had a normal, but heterogeneous, intraerythrocytic developmental cycle progression from ring to trophozoite stages at H24 (i.e., average 24% for ring stages and 76% for trophozoite stages) and from trophozoite stages to schizonts at H48 (i.e., average 26% for trophozoite stages and 74% for schizonts). However, a significant growth delay was observed for all the Cambodian CQ-R *P. falciparum* parasite lines exposed to the M_NF54_CQ20h medium. A clear shift toward younger forms was observed at H24 (i.e., average 70% for ring stages and 30% for trophozoite stages, $P < 10^{-4}$, chi-squared test) and H48 (i.e., average 8% for ring stages, 62% for trophozoite stages, and 30% for schizonts, $P < 10^{-4}$, chi-squared test) (Fig. 1). Although the intraerythrocytic developmental cycle progression was similar between the three Cambodian parasite lines at H24 in both conditions (M_NF54_CQ20h and RPMI complete medium), we observed a higher proportion of ring stages for the Cambodian *Pfkelch13* wild-type parasite line (PL3$^{WT}$) compared to the other two Cambodian *Pfkelch13* C580Y parasite lines (PL1$^{C580Y}$ and PL2$^{C580Y}$), at H48 in the M_NF54_CQ20h-treated condition (forPL3$^{WT}$: 21% ring stages, 63% trophozoite stages, and 17% schizonts; for PL1$^{C580Y}$: 0% ring stages, 58% trophozoite stages, and 42% schizonts; and for PL2$^{C580Y}$: 2% ring stages, 65% trophozoite stages, and 33% schizonts, $P < 10^{-4}$, chi-squared test).

Collectively, our data indicate that the medium prepared from CQ-killed parasites induces delayed growth of *P. falciparum* ring stages, similar to culture conditions

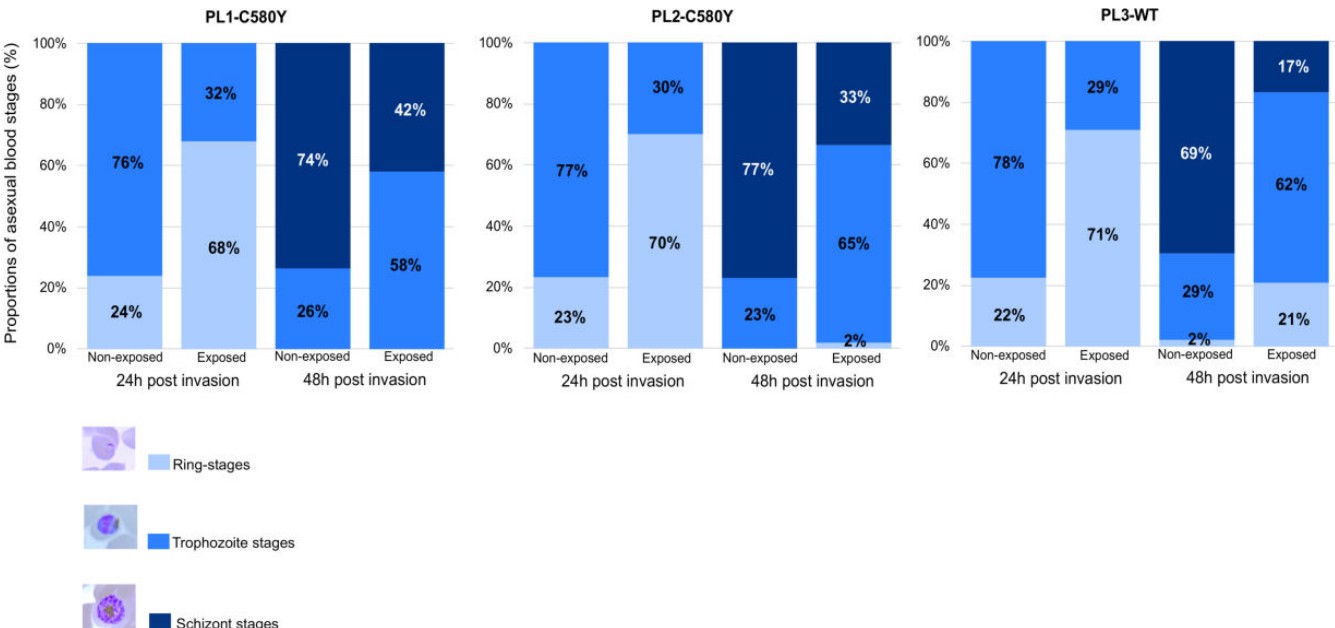

**FIG 1** Stress-induced medium mediates delayed growth of *P. falciparum* ring stages *in vitro*. Proportions of asexual blood stages (rings in light blue, trophozoites in blue, and schizonts in dark blue, Giemsa staining, microscopic examination at ×1,000 magnification) of the three Cambodian CQ-R *P. falciparum* parasite lines (PL1$^{C580Y}$, PL2$^{C580Y}$, and PL3$^{WT}$) exposed for 24 h to the M_NF54_CQ20h medium (exposed) or RPMI complete medium (non-exposed) at 24 and 48 h post-invasion are shown in each panel. For each sample, ~20,000 RBCs were assessed at 24 and 48 h for each condition to estimate the blood-stage proportions.

depleted of sugars, amino acids, lipids, and nutrients (see Discussion). This suggests that the ring-stage growth can be modulated by signals released from dead or dying parasites (i.e., soluble secreted factors or EVs). The difference in ring-stage growth retardation between parasite lines at H48 suggests that genetic background is involved in this phenotype, independent of the *Pfkech13* genotype. We also observed a marked heterogeneity in the progression of the intraerythrocytic developmental cycle of the synchronous 0–3 hpi ring-stage populations in both conditions.

### *In vitro* susceptibility to DHA is reduced in early ring-stage parasites (0–3 hpi) exposed to stress-induced medium

We next investigated whether the *in vitro* DHA susceptibility of *P. falciparum* parasite lines, as expressed by the ring-stage survival assay (RSA[0–3h]), is affected by the delayed growth progression induced by M_NF54_CQ20h treatment. Early 0–3 hpi ring-stage parasites of 3D7[C580Y] and the parasite lines PL1[C580Y] and PL2[C580Y] were prepared (Fig. S3) and exposed to four different culture media for 30 min (RPMI complete medium used as control, M_NF54_CQ20h medium prepared as previously described, M_NF54 culture medium prepared from asynchronous NF54 parasites, and M_RBC_CQ20h culture medium prepared from uninfected RBCs exposed to 200 nM CQ for 20 h). The 3D7[C580Y], PL1[C580Y], and PL2[C580Y] parasite lines were then exposed to 700 nM DHA or 0.1% dimethyl-sulfoxide (DMSO, used as a vehicle control) for 6 h, washed three times with incomplete RPMI medium to remove the drug, transferred to 6-well plates, and cultured in complete RPMI medium for a further 66 h. Parasitemia was measured by microscopy at 72 h. Parasite survival was expressed as the ratio of viable parasites in DHA to DMSO-treated samples, as previously described. All experiments were performed in triplicate.

We found that the proportions of viable parasites of the parasite lines treated with complete RPMI medium were 10.6%, 17.3%, and 18.3% for 3D7[C580Y], PL1[C580Y], and PL2[C580Y], respectively (Fig. 2). Similar proportions of viable parasites were found for parasite lines treated with culture medium prepared from asynchronous NF54 parasites (M_NF54) (10.6% for 3D7[C580Y], $P = 0.8$; 25.5% for PL1[C580Y], $P = 0.1$; and 21.3% for PL2[C580Y], $P = 0.1$, $t$-test) and parasites treated with culture medium prepared from uninfected RBCs

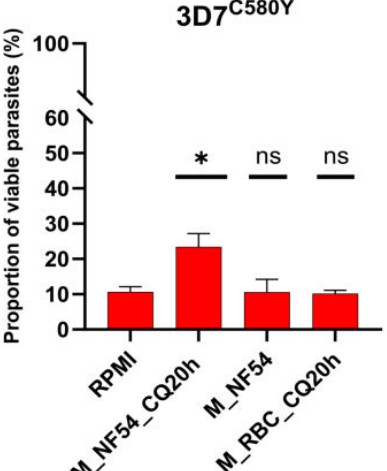 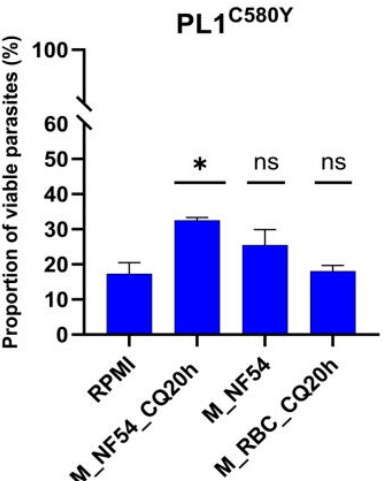 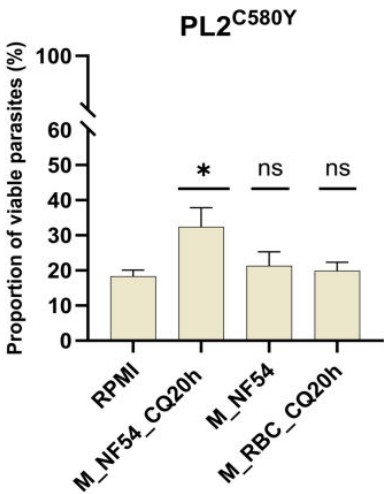

**FIG 2** *In vitro* susceptibility to DHA is reduced in early ring-stage parasites (0–3 hpi) exposed to stress-induced medium. The graphs show the proportion of viable parasites (%) as measured by the RSA[0–3h] for the parasite lines 3D7[C580Y], PL1[C580Y], and PL2[C580Y] in the four environmental conditions: 30 min 0–3 hpi ring stages pre-treatment with complete RPMI medium (RPMI), culture medium prepared from an asynchronous NF54 CQ-S parasite line exposed to 200 nM CQ for 20 h (M_NF54_CQ20h), culture medium prepared from asynchronous NF54 parasites (M_NF54), and culture medium prepared from uninfected RBCs exposed to 200 nM CQ for 20 h (M_RBC_CQ20h). The number of RBCs counted in each RSA[0–3h] assay was 20,000. All experiments were performed in triplicate. For each parasite line, no statistical difference was found between M_NF54 or M_RBC_CQ20h and RPMI conditions (indicated by "ns"), while survival rates in the M_NF54_CQ20h condition were significantly higher (~2-fold increase) compared to the RPMI condition (indicated by "*"). One-way ANOVA: 3D7[C580Y]: *F*-value =4.455, df = 3; PL1[C580Y]: *F*-value =6.268, df = 3; PL2[C580Y]: *F*-value =6.489, df = 3.

exposed to 200nM CQ for 20 h (M_RBC_CQ20h) (10.2% for the 3D7$^{C580Y}$, $P = 0.7$; 18.1% for PL1$^{C580Y}$, $P = 0.1$;and 19.9% for PL2$^{C580Y}$, $P = 0.7$, $t$-test). These data showed that (i) a culture medium containing 200nM CQ (M_RBC_CQ20h condition) did not alter the *in vitro* susceptibility of the tested parasite lines to DHA, consistent with the mode of action of the 4-aminoquinoline derivatives, which act mainly on mature stages by inhibiting the conversion of heme to hemozoin, and (ii) culture medium prepared from an asynchronous NF54 parasite line (M_NF54 condition) did not result in medium depletion or release of signals that could affect the growth progression of the ring-stages. However, we clearly observed that the proportion of viable parasites in the parasite lines treated with M_NF54_CQ20h medium was significantly increased (~2-fold) compared to the tested parasite lines exposed to the complete RPMI medium. Parasite survival changed from 10.6% to 23.4%, $P = 0.01$ for 3D7$^{C580Y}$, 17.3% to 32.5% for PL1$^{C580Y}$, $P = 0.02$, and 18.3% to 32.4% for PL2$^{C580Y}$, $P = 0.02$, $t$-test. These data may indicate that CQ-killed parasites release a signal in the culture medium that delays the growth of CQ-R ring stages and increases the proportion of ring stages that survive a 700 nM pulse of DHA.

### In vitro susceptibility of *P. falciparum* parasite lines to DHA is restored by dilutions of stress-induced medium

To confirm whether a signal released by CQ-killed parasites in the culture medium could affect the growth of CQ-R ring-stage parasite lines, we treated 0–3 hpi ring-stage 3D7$^{C580Y}$, PL1$^{C580Y}$, and PL2$^{C580Y}$ parasites with dilutions of the M_NF54_CQ20h medium (1/2 and 1/4 dilutions) before assessing their *in vitro* susceptibility to DHA using the RSA$^{0-3h}$ (Fig. S4). RPMI medium and the pure M_NF54_CQ20h medium were used as negative and positive controls, respectively. All assays were performed in triplicate.

As previously observed, we confirmed that the proportions of viable parasites of the parasite lines treated with the M_NF54_CQ20h medium were significantly increased compared to the tested parasite lines exposed to complete RPMI medium. We observed that the parasite survival changed from 10.7% to 29.2%, $P = 0.002$ for 3D7$^{C580Y}$; from 11.8% to 46.7% for PL1$^{C580Y}$, $P = 0.0003$; and from 18.8% to 60.2% for PL2$^{C580Y}$, $P = 0.002$, $t$-test (Fig. 3). We detected significant differences in the proportions of viable parasites in the lines that were treated with the pure M_NF54_CQ20h medium compared to the diluted M_NF54_CQ20h media. These data suggest that a signal released by CQ-killed parasites may diffuse into the culture medium and affect the ability of the ring stages to survive DHA exposure in a threshold-dependent manner. This raises the question of what molecule(s) might be responsible for such a phenotype.

### Ring-stage growth delay and reduced *in vitro* susceptibility to DHA induced by stress-induced medium are mediated by molecules of different molecular weight, regardless of the *Pfkelch13* genotype

Next, we determined the molecular weight range of the molecule(s) contained in the M_NF54_CQ20h medium that were likely to be sensed by the ring-stage parasites. We used a Centricon centrifugal filter device and Ultracel membranes with different cutoffs to exclude molecules with molecular weights <100 kDa (10 nm pore size), <30 kDa (3 nm pore size), <10 kDa (1 nm pore size), and <3 kDa (0.3 nm pore size) from the M_NF54_CQ20h medium (Fig. S5). The M_NF54_CQ20h media were prepared, as previously described, and then filtered using Centricon equipment to obtain four M_NF54_CQ20h medium filtrates (<100, <30, <10, and <3 kDa), which were later placed in 50 mL Falcon tubes. The early 0–3 hpi ring stages of the PL2$^{C580Y}$ parasite line were then exposed to the complete RPMI medium, unfiltered M_NF54_CQ20h (control), and each of the four filtrates for 30 min before assessing their *in vitro* susceptibility to DHA using the RSA$^{0-3h}$. All experiments were carried out in triplicate.

As previously observed, we confirmed that the proportion of viable parasites of the PL2$^{C580Y}$ parasite line treated with the M_NF54_CQ20h medium was significantly increased compared to the PL2$^{C580Y}$ parasite line incubated with the complete RPMI medium, with parasite survival rates increasing from 4.7% to 29.3% ($P = 0.0002$, $t$-test).

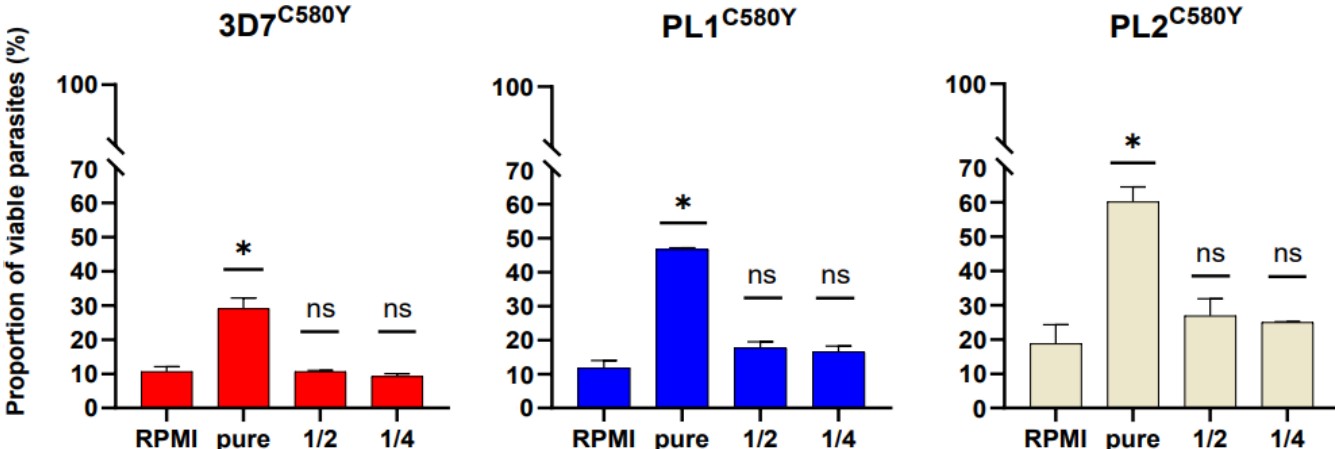

**FIG 3** *In vitro* susceptibility of *P. falciparum* parasite lines to DHA is restored by dilutions of stress-induced medium. The graphs show the proportion of viable parasites (%) as expressed by the RSA$^{0-3h}$ for the parasite lines 3D7$^{C580Y}$, PL1$^{C580Y}$, and PL2$^{C580Y}$ in the four experimental conditions: 30 min 0–3 hpi ring stages pre-treated with complete RPMI medium (negative control), pure M_NF54_CQ20h medium, 1/2 dilution of M_NF54_CQ20h medium in complete RPMI medium, and 1/4 dilution of M_NF54_CQ20h medium in complete RPMI medium. The number of RBCs counted in each RSA$^{0-3h}$ assay was ~20,000. All experiments were performed in triplicate. For each parasite line, no statistical difference was found between RPMI and 1/2 and 1/4 M_NF54_CQ20h (indicated by "ns"), while survival rates in the pure M_NF54_CQ20h condition were significantly higher compared to the RPMI condition (indicated by "*"). One-way ANOVA: 3D7$^{C580Y}$: *F*-value =9.031, df = 3; PL1$^{C580Y}$: *F*-value =84.42, df = 3; PL2$^{C580Y}$: *F*-value =26.86, df = 3.

We found similar proportions of viable parasites of the PL2$^{C580Y}$ parasite line treated with pure unfiltered M_NF54_CQ20h, <100 and <30 kDa M_NF54_CQ20h filtrates (<100 kDa filter, 28.3%, $P = 0.63$; <30 kDa filter, 24.4%, $P = 0.06$, *t*-test), whereas the proportion of viable parasites of the PL2$^{C580Y}$ parasite line treated with <10 and <3 kDa M_NF54_CQ20h filtrates was slightly reduced (<10 kDa filter, 15.4%, $P = 0.0016$; <3 kDa filter, 17.3%, $P = 0.003$, *t*-test). The reduction in the proportion of viable parasites treated with <10 and <3 kDa M_NF54_CQ20h filtrates remained significantly higher than the proportion detected in the RPMI medium control condition (<10 kDa filter, $P = 0.0001$ and <3 kDa filter, $P = 0.0001$, *t*-test) (Fig. 4A). These results suggest that the signal involved in the delayed growth progression of the ring stages is mediated by multiple molecules with different molecular weights (molecules >10 and <10 kDa) and argue against the possibility that the signal is mediated by EVs as their sizes range from 30 to 500 nm (Fig. 4A).

To confirm whether small molecules <3 kDa could have an effect on the ring-stage growth progression and their *in vitro* susceptibility to DHA, we exposed the PL2$^{C580Y}$ and PL3$^{WT}$ early 0–3 hpi ring stages to the complete RPMI medium (control) and the <3 kDa M_NF54_CQ20h medium filtrate for 30 min before assessing their *in vitro* susceptibility to DHA using the RSA$^{0-3h}$. We compared the PL2$^{C580Y}$ and PL3$^{WT}$ isolates to assess the effect of the *Pfkelch13* mutation, as these parasite lines from Pailin (Cambodia) have a similar genetic background (see Table S1). Both the proportions of viable PL2$^{C580Y}$ and PL3$^{WT}$ parasites treated with the <3 kDa M_NF54_CQ20h medium filtrates were significantly increased compared to parasites treated with complete RPMI medium. Parasite survival increased from 6.8% to 21.5%, $P = 0.004$, for the PL2$^{C580Y}$ parasite line and from 2.0% to 10.1%, $P = 0.0003$, for the PL3$^{WT}$ parasite line, *t*-test (Fig. 4B). We confirmed that, regardless of the *Pfkelch13* genotype, molecules <3 kDa can promote delayed ring-stage growth and reduced *in vitro* susceptibility to DHA.

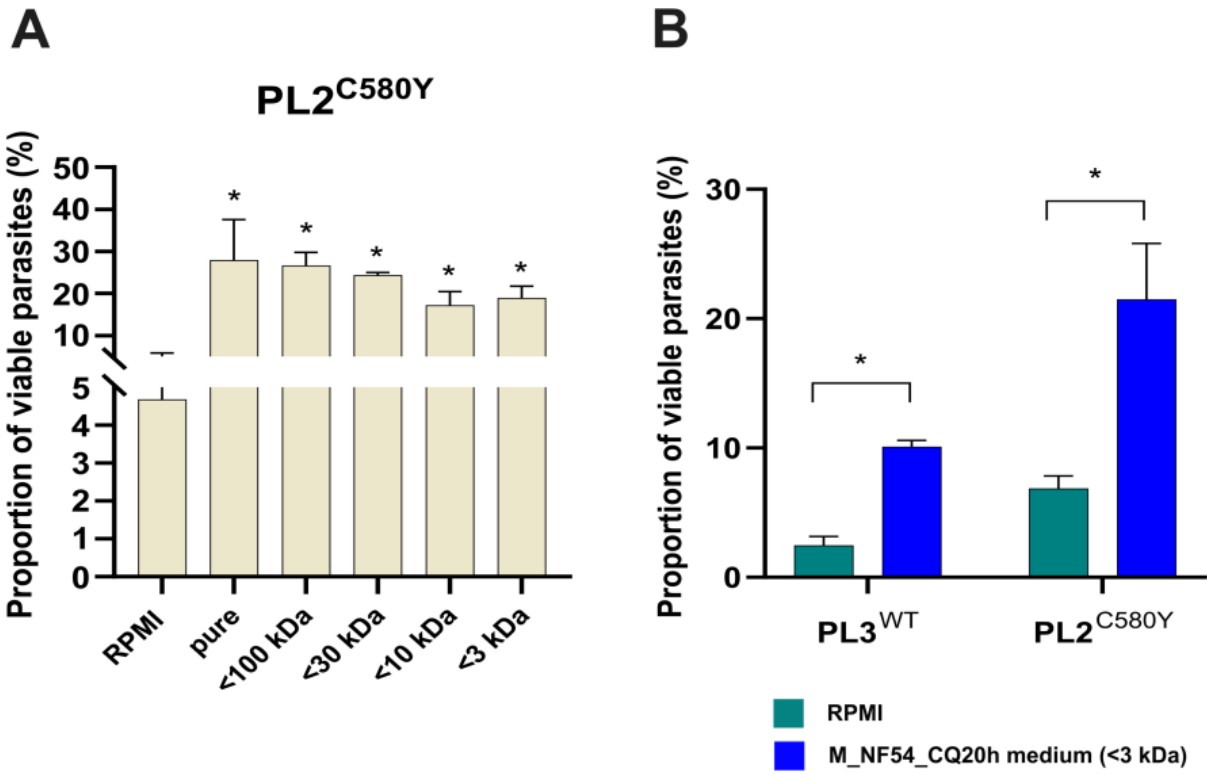

**FIG 4** Ring-stage growth delay and reduced *in vitro* susceptibility to DHA induced by stress-induced medium are mediated by molecules of different molecular weights, regardless of the *Pfkelch13* genotype. (A) The graph shows the proportion of viable parasites (%) as expressed by the RSA$^{0-3h}$ for the PL2$^{C580Y}$ line in the six experimental conditions: 30 min 0–3 hpi ring stages pre-treated with complete RPMI medium (RPMI, negative control), pure unfiltered M_NF54_CQ20h (M_NF54_CQ20h medium, positive control), and four M_NF54_CQ20h medium filtrates (<100, <30, <10, and <3 kDa). The number of RBCs counted in each RSA$^{0-3h}$ assay was ~20,000. All experiments were performed in triplicate. Survival rates in the pure M_NF54_CQ20h condition and all M_NF54_CQ20h medium filtrates (<100, <30, <10, and <3 kDa) were significantly higher compared to the RPMI condition (indicated by "*"). PL2$^{C580Y}$: *F*-value =7.672, df = 3. (B) The graph shows the proportion of viable parasites (%) as measured by the RSA$^{0-3h}$ for the PL2$^{C580Y}$ and PL3$^{WT}$ lines in two experimental conditions: 30 min 0–3 hpi ring stages pre-treated with complete RPMI medium (negative control) and the <3 kDa M_NF54_CQ20h medium filtrate. The number of RBCs counted in each RSA$^{0-3h}$ assay was ~20,000. All experiments were performed in triplicate. For both parasite lines (PL2$^{C580Y}$ and PL3$^{WT}$), survival rates were significantly higher in the <3 kDa M_NF54_CQ20h medium filtrate condition compared to the RPMI condition (indicated by "*"). *t*-test summary: PL2$^{C580Y}$: *t* = 6.450, df = 2; PL3$^{WT}$: *t* = 22.10, df = 2.

## Mature-stage parasites treated with DHA release molecules into the culture medium that reduce the *in vitro* susceptibility of the ring-stage parasites to DHA

We then tested whether a co-culture of *P. falciparum* ring and mature stages exposed to 700 nM DHA for 6 h could promote delayed *P. falciparum* ring-stage growth and reduced *in vitro* susceptibility to DHA. Ring and mature stages were cultured in the same medium, physically separated by a polycarbonate membrane (upper chamber and lower compartment), using a Transwell insert system (0.4 µm pore size). We first exposed the parasite lines (ring or mature stages) or uninfected RBCs to 700 nM DHA or DMSO (0.1%) for 30 min in the upper chamber. We then seeded early 0–3 hpi ring stages into the lower compartment that were then cultured for 6 h in the same culture medium, allowing DHA and molecules released from the upper compartment to passively diffuse into the culture medium. Parasites in the lower compartment were collected, washed twice in an incomplete RPMI medium to remove the drug, transferred into 6-well plates, and cultured in complete RPMI medium for a further 66 h. Parasitemia was measured by microscopy after 72 h. Parasite survival was expressed as the ratio of viable parasites in DHA to DMSO-treated samples, as previously described. All the experiments were performed in triplicate.

We used the 3D7^WT and the 3D7^C580Y parasite lines in the four experimental conditions (upper/lower compartments): 22–36 hpi mature stages/early 0–3 hpi ring stages, early 0–3 hpi ring stages/early 0–3 hpi ring stages, uninfected RBCs/early 0–3 hpi ring stages, and uninfected RBCs/22–36 hpi mature stages (Fig. S6). The proportions of viable parasites when 3D7^WT and 3D7^C580Y early 0–3 hpi ring stages were co-cultured with uninfected RBCs were 0.5% and 3.8%, respectively (Fig. 5A). When the 3D7^WT and 3D7^C580Y mature stages were co-cultured with uninfected RBCs, the proportions of viable parasites were 0.4% and 0.4%, respectively. A significant difference in *in vitro* susceptibility to DHA was observed between 3D7^C580Y ring stages and mature stages. This result is consistent with previous studies reporting that the mature stages of the *Pfkelch13* mutant parasites are highly susceptible to DHA (22). We also observed that DHA remains active when incubated with uninfected RBCs. As expected, we observed similar parasite survival rates (compared to the uninfected RBCs/early 0–3 hpi ring-stage condition) when 3D7^WT and 3D7^C580Y early 0–3 hpi ring stages were co-cultured with early 0–3 hpi ring stages (0.7% vs 0.5%, *P* = 0.34, for the 3D7^WT parasite line and 3.2% vs 3.8%, *P* = 0.56, for the 3D7^C580Y parasite line, *t*-test). However, when early 0–3 hpi ring stages of 3D7^WT and 3D7^C580Y were co-cultured with 22–36 hpi mature stages, a significant decrease in *in vitro* susceptibility to DHA was observed for both parasite lines compared to the control condition (early 0–3 hpi ring stages/early 0–3 hpi ring stages). Parasite survival rates increased from 0.5% to 2.8% (*P* = 0.003, *t*-test) for the parasite line 3D7^WT and from 3.8% to 13.3% (*P* = 0.003, *t*-test) for the parasite line 3D7^C580Y (Fig. 5A).

To exclude that the decrease in *in vitro* DHA susceptibility observed in the early 0–3 hpi ring-stage/22–36 hpi mature-stage co-cultures was due to a reduced concentration

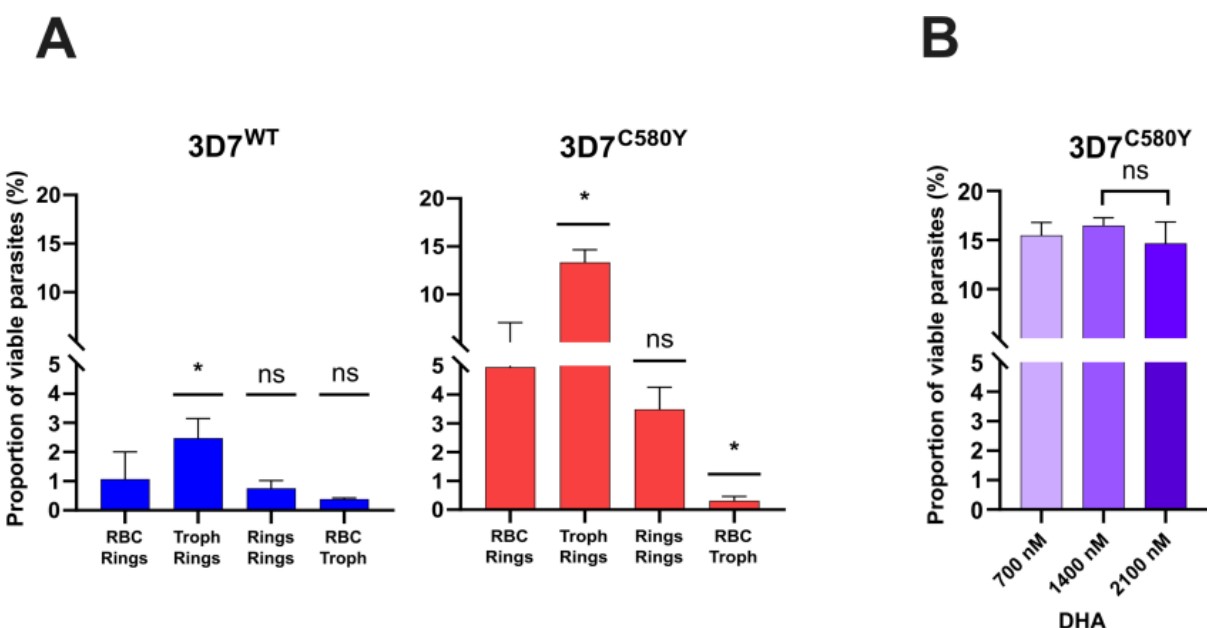

**FIG 5** Mature-stage parasites treated with DHA release molecules into the culture medium that reduce *in vitro* ring-stage susceptibility to DHA. (A) The graphs show the proportion of viable parasites (%) as expressed by the RSA^0–3h of 3D7^WT and 3D7^C580Y of 0–3 hpi ring stages co-cultured with uninfected RBCs treated with DHA (at 700 nM for 30 min), 22–36 hpi mature stages, and 0–3 hpi ring stages plus 22–36 hpi mature stages co-cultured with DHA-treated uninfected RBCs. The number of RBCs counted in each RSA^0–3h assay was ~20,000. All experiments were performed in triplicate. For both parasite lines, 0–3 hpi ring stages co-cultured with 22–36 hpi mature stages showed a significant decrease in *in vitro* ring-stage susceptibility to DHA compared to the control condition (uninfected RBCs/early 0–3 hpi rings). No statistical difference was found in the other conditions, except for 3D7^C580Y mature stages co-cultured with uninfected RBCs, which showed a significant reduction in the proportion of viable parasites compared to the control condition (uninfected RBCs/early 0–3 hpi rings). One-way ANOVA: 3D7^WT: *F*-value =7.140, df = 3; 3D7^C580Y: *F*-value =53.26, df = 3. (B) The graph shows the proportion of viable parasites (%) as measured by the RSA^0–3h of 3D7^C580Y 0–3 hpi ring stages co-cultured with 22–36 hpi mature stages treated with DHA (700, 1,400, and 2,100 nM for 30 min). The number of RBCs counted in each RSA^0–3h assay was ~20,000. All experiments were performed in triplicate. The proportions of viable parasites were found similar regardless of DHA concentration. One-way ANOVA: 3D7^C580Y: *F*-value =1.022, df = 2.

of DHA available in the medium as a result of DHA "consumption" by the mature stages, we performed additional early 0–3 hpi ring-stage/22–36 hpi mature-stage co-cultures with the 3D7$^{C580Y}$ parasite line in which DHA concentrations were increased up to 1,400 and 2,100 nM (Fig. 5B). We observed similar proportions of viable parasites at all concentrations (15.5% at 700 nM DHA vs 16.5% at 1,400 nM DHA, $P = 0.3$, and 14.6% at 2,100 nM DHA, $P = 0.6$), ruling out the hypothesis of reduced DHA availability. These findings suggest that drug-exposed susceptible mature stages of *P. falciparum* may release molecules as they die, which are then used for population-level communication. These potential molecules could thus act as danger signals of environmental stress and induce temporary growth arrest of the ring stages to optimize population survival.

## DISCUSSION

In this study, we report evidence that the stress-induced extracellular environment can elicit a reversible growth arrest phenotype in multiple *P. falciparum* parasite lines, independent of the *Pfkelch13* genotype (Fig. 1). Furthermore, we show that *in vitro* susceptibility to DHA can vary in a threshold-dependent manner when 0–3 hpi ring stages are exposed to a stress-induced medium (M_NF54_CQ20H), regardless of their *Pfkelch13* genotype (Fig. 2 and 3). This suggests the existence of a specific signaling threshold to induce ring-stage growth arrest. According to our results (Fig. 4), we have identified these extracellular signals as a potential group of molecules <3 kDa. The signal appeared to be released from dying trophozoites (Fig. 5), as shown in our final experiment.

These findings are in line with previous studies that have reported on the ability of *P. falciparum* to sense extracellular signals. For example, lysophosphatidylcholine, a compound found in the human host, was shown to induce sexual commitment in blood-stage parasites (35). Other reports have suggested the ability of *P. falciparum* to monitor its population with a density-sensing mechanism and promote apoptosis if the parasitemia exceeds a certain threshold (36, 37). Moreover, several polypeptides appear to be produced and secreted by parasites to regulate the cell cycle in other models, such as the kinetoplastid *Trypanosoma brucei* (38). Cell growth arrest or dormancy phenotypes have been rescued by amino acid deprivation (30), artemisinin exposure (29, 33), and abiotic stress like heat (31, 32, 39). In addition, growth arrest can be shortened by exposing the parasite to phytohormones such as abscisic acid and gibberellic acid *in vitro* (40). Thus, cell growth arrest appears to be a common response when the parasite is exposed to adverse conditions. Although the composition of the stress-induced M_NF54_CQ20h medium remains unknown, we can make some hypotheses about the growth-arresting factor(s).

Our results are unlikely to involve lysophosphatidylcholine as a cell growth arrest inducer since this molecule is of human host origin (35). Instead, our data suggest that the signal originates from the parasite itself, or more specifically, from the dying trophozoite (Fig. 5). Interestingly, EVs appear to not be involved in cell growth arrest signaling, although they have been described as a host–parasite communication mechanism, influencing immune response regulation and cargo delivery (41–46). Indeed, EVs have been reported in the literature to range in size from 30 to 500 nm (47), whereas the less stringent filter we used allows only 10 nm-wide particles to pass (Fig. 4; Fig. S5). However, as the signaling molecules were not identified, the results from the size exclusion analysis could also be explained by a single small molecule present as monomers and polymers or aggregated. Similarly, the candidate molecule may be a larger molecule that degrades while the smaller products still retain activity.

Thus, the parasite appears to rely on several different mechanisms for extracellular communication, involving both EVs and small released molecules. Strikingly, our data showed that ring-stage growth arrest reduces artemisinin susceptibility, irrespective of the *Pfkelch13* genotype (Fig. 2 to 5). Our hypothesis is that ring-stage growth arrest targets the hemoglobin degradation pathway in a similar way to *Pfkelch13* mutations conferring artemisinin resistance. Indeed, we recovered an average of 70% of the ring

stage 24 h after exposure to M_NF54_CQ20h, regardless of *Pfkelch13* genotype. This means that these parasites have significantly reduced their cycle progression and, thus, hemoglobin degradation as most parasites are expected to be in the trophozoite stage under normal conditions (Fig. 1). On the other hand, previous reports have shown a reduced uptake of hemoglobin from the host cell in artemisinin-resistant parasites with the *Pfkelch13* mutation (25, 26). This suggests that growth arrest is a constitutive survival mechanism in the early ring stage, triggered by specific unknown factors released by trophozoites. We propose that ring-stage growth arrest and artemisinin resistance mediated by *Pfkelch13* mutations are two independent, but complementary, resistance mechanisms that efficiently reduce *in vitro* susceptibility to DHA in the early ring stage.

By adding the extracellular environment as an essential component of ring-stage growth arrest and parasite survival upon DHA exposure, this study provides new insights into artemisinin resistance. Further research is needed to unravel the ring-stage growth arrest signaling pathway and the nature of the extracellular factors released. For example, we do not know how the extracellular signals reach their target. However, the signals may interact with a surface receptor, as such a mechanism was already described (48), or pass through the erythrocyte membrane in an active (transporter -mediated) or passive manner. This work may provide a foundation for the development of a novel therapeutic approach based on quorum quenching, which consists of disrupting extracellular communication. Preventing ring-stage growth arrest by disrupting extracellular signaling could restore the susceptibility of *P. falciparum* parasites to artemisinin. An untargeted metabolomics analysis should be performed to unravel the nature of the stress signal detected in our experiments.

In conclusion, this study demonstrates that an unknown extracellular signal <3 kDa released from dying trophozoites can induce in a threshold-dependant manner a temporary growth arrest in the early ring stage of *P. falciparum*, thereby reducing its susceptibility to artemisinin, independent of the *Pfkelch13* genotype.

## MATERIALS AND METHODS

### *P. falciparum* parasite *in vitro* culture

*P. falciparum* asexual blood-stage parasites were propagated in T50 flasks (Falcon) at 4% hematocrit and parasitemia <1% in 10 mL RPMI-1640 medium (Gibco) supplemented with 10% Albumax (Gibco), 0.2 mM hypoxanthine, 2% AB human serum (heat-inactivated at 56°C), and 4 µg/mL gentamicin (complete RPMI medium). Parasites were maintained at 37°C in 90% $N_2$, 5% $O_2$, and 5% $CO_2$. $AB^+$ human sera and fresh $O^+$ blood from healthy donors were provided by the Établissement Français du Sang at Strasbourg. We used several parasite lines including the laboratory strains 3D7 and NF54, 3D7$^{C580Y}$ (a CRISPR/Cas9 edited line (49), and three culture-adapted isolates from Pailin (western Cambodia). These include PL1$^{C580Y}$ and PL2$^{C580Y}$ (two *Pfkelch13 580Y* mutant parasite lines) and PL3$^{WT}$ (a *Pfkelch13* wild-type parasite line). The genotypes of the parasite lines in the *P. falciparum chloroquine resistance transporter* gene (*pfcrt*), the *Plasmodium falciparum multidrug-resistant protein* gene (*pfmdr-1*), and the *Plasmodium falciparum kelch* gene located on chromosome 13 (*Pfkelch13*) were determined by whole-genome sequencing. Details are presented in Table S1.

### Parasite synchronization and ring-stage survival assay (RSA$^{0–3h}$)

Parasites were cultured until parasitemia was between 1% and 1.5%. Synchronized parasite cultures were obtained by exposing ring-stage parasites to 5% D-sorbitol (Sigma) for 10 min at 37°C to remove mature-stage parasites, as previously described (50). The kinase inhibitor ML10 (200 nM) was added to the culture medium at 20 h post-sorbitol treatment (trophozoite stages) to block schizont rupture as previously reported (51, 52). After 37–49 h in culture, parasite stages were checked on Giemsa-stained blood smears. Multinucleated schizonts were collected by differential

centrifugation (8 min, 600 $g$, no brake) over a density gradient of 75% Percoll (Sigma). Schizonts were resuspended in 10 mL of the complete RPMI culture medium with 200 µL RBCs and centrifuged at 2,000 rpm for 5 min to remove ML10 and residual Percoll. The RBC pellet was resuspended in 10 mL of complete RPMI culture medium. The cultures were incubated at 37°C in 90% $N_2$, 5% $O_2$, and 5% $CO_2$ on a shaker plate for 3 h. Early rings (0–3 hpi) were treated with 5% D-sorbitol to remove residual schizonts. Blood-stage parasites and parasitemia were checked on Giemsa-stained blood smears, and parasitemia was adjusted to 0.5%.

*In vitro* RSA$^{0–3h}$ was performed as previously described (22). Briefly, tightly synchronized 0–3 hpi early ring stages were exposed to a pharmacologically relevant dose of 700 nM DHA or 0.1% DMSO (vehicle control) for 6 h at 0.5% parasitemia and 4% hematocrit, washed twice with RPMI incomplete medium to remove the drug, transferred to 6-well or 12-well plates, and cultured for an additional 66 h in a drug-free medium at 4% hematocrit. RBC pellets were then collected to prepare Giemsa-stained blood smears for estimation of parasitemia. The number of RBCs counted in each RSA$^{0–3h}$ assay was ~20,000. Parasite survival was expressed as the percentage value of the parasitemia in DHA-treated samples divided by the parasitemia in DMSO-treated samples processed in parallel.

## Stress induced and supernatant prepared M_NF54_CQ20h

*P. falciparum* NF54 (chloroquine-sensitive) parasites were cultured and maintained asynchronously. When 1% parasitemia was reached, infected RBCs were treated with chloroquine (Sigma) at a final concentration of 200 nM for 20 h, and the culture was incubated at 37°C, 90% $N_2$, 5% $CO_2$, and 5% $O_2$. The culture was then centrifuged (2,000 rpm, 5 min, room temperature), and the culture medium (i.e., the supernatant) was collected in a 50 mL Falcon tube and stored at 37°C in a water bath until use. The supernatant was not filtered, in order to retain all components and molecules in the chloroquine-induced stress medium culture. Supernatants were prepared extemporaneously for each experiment. Chloroquine-induced stress in the NF54 parasite line was assessed by Giemsa-stained blood smears before and after chloroquine exposure (see Supplemental Information—1. Stress-induced medium mediates delayed growth of *P. falciparum* ring stages *in vitro*).

## Statistical analysis

Data were analyzed with Microsoft Excel, MedCalc version 20 (Mariakerke, Belgium), and GraphPad Prism (v.9.0.0). Quantitative data were expressed as the mean (±SEM). Continuous variables were compared with the one-sample *t*-test. Associations were analyzed with the Spearman correlation test. We deemed significant *P*-values of less than 0.05.

## ACKNOWLEDGMENTS

We thank the staff of the ICAReB platform at Institut Pasteur for providing blood samples from healthy volunteers. We are also grateful to the BIHP staff (Biologie des Infections Hôtes-Parasites) unit at Institut Pasteur for their advice and technical support. We deeply thank Professor David Baker from the London School of Hygiene and Tropical Medicine for providing ML10. We acknowledge the help of the HPC Core Facility of the Institut Pasteur for this work as well as the Biomics Platform, C2RT, Institut Pasteur, Paris, France (supported by France Génomique, ANR-10-INBS-09 and IBISA).

This work was supported by the Institut Pasteur, Paris, the French Government (Agence Nationale de la Recherche), Laboratoire d'Excellence (LabEx) "French Parasitology Alliance for Health Care" (ANR-11-15 LABX-0024-PARAFRAP), and the University of Strasbourg through the Programme IdEX 2022 to D.M. L.P. has received financial support from the Sorbonne Université, Collège Doctoral ED 515 Complexité du Vivant. D.A.F. gratefully acknowledges funding support from the NIH (R01 AI109023).

L.P. and D.M. designed the experiments. L.P. carried out *in vitro* cultures, drug testing, and molecular biology testing and analysis. L.P., D.L., D.A.F., and D.M. conducted the data analysis. D.L. and D.A.F. provided a critical review of the data and provided experimental guidance. D.L., D.A.F., and D.M. provided critical edits to the manuscript. All authors read and approved the final manuscript.

## AUTHOR AFFILIATIONS

[1]Malaria Genetics and Resistance Unit, INSERM U1201, Institut Pasteur, Université Paris Cité, Paris, France

[2]Sorbonne Université, Collège Doctoral ED 515 Complexité du Vivant, Paris, France

[3]Malaria Parasite Biology and Vaccines Unit, Institut Pasteur, Université Paris Cité, Paris, France

[4]Institute of Parasitology and Tropical Diseases, UR7292 Dynamics of Host–Pathogen Interactions, Université de Strasbourg, Strasbourg, France

[5]Department of Drug Discovery, Medicines for Malaria Venture, Geneva, Switzerland

[6]Department of Microbiology and Immunology, Columbia University Irving Medical Center, New York, New York, USA

[7]Center for Malaria Therapeutics and Antimicrobial Resistance, Division of Infectious Diseases, Department of Medicine, Columbia University Irving Medical Center, New York, New York, USA

[8]Laboratory of Parasitology and Medical Mycology, CHU Strasbourg, Strasbourg, France

## PRESENT ADDRESS

Lucien Platon, Institute of Parasitology and Tropical Diseases, UR7292 Dynamics of Host-Pathogen Interactions, Université de Strasbourg, Strasbourg, France

Didier Ménard, Institute of Parasitology and Tropical Diseases, UR7292 Dynamics of Host-Pathogen Interactions, Université de Strasbourg, Strasbourg, France

## AUTHOR ORCIDs

Lucien Platon http://orcid.org/0000-0001-7894-5977
David A. Fidock http://orcid.org/0000-0001-6753-8938
Didier Ménard http://orcid.org/0000-0003-1357-4495

## FUNDING

| Funder | Grant(s) | Author(s) |
| --- | --- | --- |
| Institut Pasteur | | Didier Ménard |
| Agence Nationale de la Recherche (ANR) | ANR-11-15 LABX-0024-PARAFRAP | Didier Ménard |
| Université de Strasbourg (University of Strasbourg) | Programme IdEX 2022 | Didier Ménard |
| Sorbonne Université (Sorbonne University) | Collège Doctoral ED 515 | Lucien Platon |
| National Institute of Allergy and Infectious Diseases/National Institutes of Health (NIAID/NIH) | R01 AI109023 | David A. Fidock |
| Hôpitaux Universitaires de Strasbourg (Direction la Recherche Clinique et des Innovations) | | Didier Ménard |

## AUTHOR CONTRIBUTIONS

Lucien Platon, Conceptualization, Formal analysis, Methodology, Validation, Writing – original draft | Didier Leroy, Formal analysis, Methodology, Validation, Visualization,

Writing – review and editing | David A. Fidock, Formal analysis, Methodology, Valida-
tion, Visualization, Writing – review and editing | Didier Ménard, Conceptualization,
Data curation, Formal analysis, Funding acquisition, Investigation, Methodology, Project
administration, Resources, Supervision, Validation, Writing – review and editing

## ADDITIONAL FILES

The following material is available online.

### Supplemental Material

**Supplemental information (Spectrum03500-23-S0001.docx).** Supplemental table and
figures.

### Open Peer Review

**PEER REVIEW HISTORY (review-history.pdf).** An accounting of the reviewer comments
and feedback.

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
