## [Reviewer comments · Microbiology Spectrum]

Microbiology Spectrum

Drug-induced stress mediates *Plasmodium falciparum* ring-stage growth arrest and reduces in vitro parasite susceptibility to artemisinin

Lucien Platon, Didier Leroy, David Fidock, and Didier Menard

Corresponding Author(s): Didier Menard, Universite de Strasbourg

Review Timeline:

Submission Date:	September 27, 2023
Editorial Decision:	October 31, 2023
Revision Received:	December 14, 2023
Accepted:	January 15, 2024

Editor: Anat Florentin

Reviewer(s): The reviewers have opted to remain anonymous.

Transaction Report:

DOI: <https://doi.org/10.1128/spectrum.03500-23>

Re: Spectrum03500-23 (Drug-induced stress mediates Plasmodium falciparum ring-stage growth arrest and reduces in vitro parasite susceptibility to artemisinin)

Dear Dr. Didier Menard:

Thank you for the privilege of reviewing your work. Below you will find my comments, instructions from the Spectrum editorial office, and the reviewer comments.

Revision Guidelines

Sincerely,
Anat Florentin
Editor
Microbiology Spectrum

Reviewer #1 (Comments for the Author):

This is an interesting study towards identifying factors released by parasites that affect the survival of the ring stage of the parasite. The study is well designed and it answers some of the important questions. I have a few minor comments: Figure 1 shows the proportion of different stages of the parasite but it does not indicate what was the parasitemia. Given that higher parasitemia itself can cause stress, it would be good to mention what was the total parasitemia across all experimental

conditions.

Labelling of supplementary information is not correct. It seems Figure S2 is added at the last stage and not referred to in the text. Page 5 line 135 should be S1 and S2. Page 6 line 173 should be S3 and so on. These need to be corrected.

Page 7 line 210 mentions 3/4 dilution of media. Data for this is not included in Figure 3. Figure S4 (which is referred to as S3) also does not show 3/4 dilution.

It seems 'M_NF54_CQ20h' in Figure 2 and 'pure' in Figure 3 refer to the same culture media. I am unsure why the percentage survival is different in M_NF54_CQ20h' Vs 'pure'. If these are two different things then it should be explained more clearly.

The experiment shown in Figure 5 is complex. The inclusion of schematics as part of Figure 5 will help understand the experiment.

An explanation of why PL3WT was used in the Figure 4 study would help the reader understand this better (page 8, line 260).

Do authors think that the use of membrane filters may have excluded some of the hydrophobic components in media which could have affected ring stage parasite survival? An explanation or a justification would be helpful.

Otherwise, it is a nice study and it is well presented.

Reviewer #2 (Comments for the Author):

The study by Paton et al. presents an interesting new insight on ART-Pf interactions with potential clinical significance. The results clearly indicate that the parasite's response to ART can be modified by exposure to conditioned culture medium prepared by killing parasites with CQ and collecting the supernatant ('death signal'). The authors suggest the death signal released by dying trophozoites is a cell-cell signaling mechanism that leads to enhanced ring-stage survival after ART exposure independent of pfk13 genotype. The possibility that this signal is mediated EVs was eliminated size-exclusion spin columns and the retained potency of different size-exclusion fractions was interpreted by the authors to indicate the death signal has multiple chemical mediators of different sizes. These have potential important implications on evolving multigenic ART-R mechanisms as such a feedback loop independent of K13 could be a driver for selection of other types of parasite tolerance to ART exposure. This interpretation is consistent with other recent studies that have demonstrated a clear link between induction of parasite stress responses and altered sensitivity to ART. Overall this is a superb study that adds to our understanding of factors influencing parasite responses to antimalarial drugs.

Major concern:

What is not clear from the study is the origin and nature of the active ingredient(s) of the death signal. The description of how the supernatant was prepared is inadequate. It is essential to include a detailed protocol in the Methods or as a separate supplement, so others can repeat the conditions used. This would include catalog numbers, etc... Include whether the supernatant was filtered or centrifuged to remove membrane fragments.

Minor concerns:

1. It would be useful to include counts on gametocyte abundance in addition to asexual stages.
2. Provide the rationale how was it decided on the stress-induced medium conditions used in this study.
3. What was RBC phenotype of blood used for culture, and sera?
4. Since the effector molecules were not identified, the results from the size exclusion analysis could also be explained by a single small molecule present as monomers and polymers or aggregated. Similarly, the candidate molecule may be a larger molecule that degrades while the smaller products still retain activity. Such alternatives should be included in the discussion. Also, the authors cannot exclude the possibility that the active molecules are released from modified RBC of the of parasite-infected cells.
5. L317 - "secrete" implies a specific biological process and there are data provided to support this is the active mechanism involved. "released" would be a more accurate term to use until the process is better defined. The accompanying speculation on similarities to other organisms should be moved out of the Results and into the Discussion.
6. The data presented do not adequately support interpretation of a dose-dependent response, and the authors should stop at "threshold dependent".
7. Instead of reference #36 (a review), primary references should be cited for these studies.
8. The end paragraphs are highly speculative considering no molecules mediating the altered responses were identified, no mechanism was identified, nor were any critical metabolic pathways defined. These paragraphs should be deleted or significantly revised without the speculation.

Malaria Parasite Biology and Vaccines
Department of Parasites and Insect Vectors
Institut Pasteur
25-28 Rue du Dr Roux
75724 Paris Cedex 15

**Institut
de parasitologie et
de pathologie
tropicale**

Dr Anat Florentin, Editor
Microbiology Spectrum

SPECTRUM03500-23

Platon L, et al. Drug-induced stress mediates Plasmodium falciparum ring-stage growth arrest and reduces in vitro parasite susceptibility to artemisinin.

We thank you for the detailed reviews of our manuscript and for your editorial comments and assessment, which we have carefully read. We feel we have been able to thoroughly address all comments, which are incorporated in our revised manuscript.

The external reviewers make a series of excellent and detailed recommendations that have been incorporated into our revised manuscript, as detailed below in our point-by-point reply.

We have also carefully edited our manuscript to make the report more concise and accessible to the broad research readership of *Microbiology Spectrum*. The revised version complies with the journal's word restriction (title, main text and tables/figures) and is in compliance with the *Microbiology Spectrum* publication guidelines format.

Reviewers' Comments:

Reviewer #1 (Comments for the Author):

This is an interesting study towards identifying factors released by parasites that affect the survival of the ring stage of the parasite. The study is well designed, and it answers some of the important questions.

Reply: We appreciate this comment along with the interest and time that the reviewer #1 has taken to review our manuscript.

I have a few minor comments:

Figure 1 shows the proportion of different stages of the parasite, but it does not indicate what was the parasitemia. Given that higher parasitemia itself can cause stress, it would be good to mention what was the total parasitemia across all experimental conditions.

Reply: We have added the parasitemia in line 136 and in the S1 legend in the supplementary data, acknowledging the comment of reviewer #1. A parasitemia of 0.5% was used in all the experiments that were performed and the parasites developed normally at this cell density.

Labelling of supplementary information is not correct. It seems Figure S2 is added at the last stage and not referred to in the text. Page 5 line 135 should be S1 and S2. Page 6 line 173 should be S3 and so on. These need to be corrected.

Reply: These errors have been corrected accordingly.

Page 7 line 210 mentions 3/4 dilution of media. Data for this is not included in Figure 3. Figure S4 (which is referred to as S3) also does not show 3/4 dilution.

Reply: The reference to 3/4 dilution in the main text (line 209) has been removed.

It seems 'M_NF54_CQ20h' in Figure 2 and 'pure' in Figure 3 refer to the same culture media. I am unsure why the percentage survival is different in M_NF54_CQ20h Vs 'pure'. If these are two different things, then it should be explained more clearly.

Reply: We confirm that the supernatant "M_NF54_CQ20h" and "pure" in Figures 2 and 3 are the same culture media. "Pure" refers to the undiluted supernatant "M_NF54_CQ20h" as shown in Figure S4 and in line 210 of the main text. The RSA rates shown in Figures 1 and 2 are based on independent experiments. We agree that the differences in RSA rates are likely due to variability in the production of M_NF54_CQ20h, which we cannot be fully controlled for as we do not know the exact composition of such a supernatant. However, we observed that the trends were clearly maintained, with a visible increase in RSA rates following DHA exposure.

The experiment shown in Figure 5 is complex. The inclusion of schematics as part of Figure 5 will help understand the experiment.

Reply: The Supplementary Figure S5 contains a detailed visual schematic of the experiment shown in Figure 5.

An explanation of why PL3WT was used in the Figure 4 study would help the reader understand this better (page 8, line 260).

Reply: The parasite strain PL3^{WT} was compared with PL2^{C580Y} because of their genetic proximity. In fact, both parasites are Cambodian field isolates collected in Pailin in the same year (see Table 1 in the supplementary data). They carry a different *Pfkelch13* genotype. As suggested by the reviewer #1, we have added the following statement in the main text (line 260), which reads as follows: 'We compared the PL2^{C580Y} and PL3^{WT} field isolates to assess the effect of the *Pfkelch13* mutation, as these parasite lines from Pailin (Cambodia) have a close genetic background (see Table S1)'.

Do authors think that the use of membrane filters may have excluded some of the hydrophobic components in media which could have affected ring stage parasite survival? An explanation or a justification would be helpful.

Reply: It cannot be excluded that the use of membrane filters may have excluded some of the hydrophobic components in the media that may have affected the survival of ring stage parasites, as suggested by reviewer #1. However, we observed the same phenotypes and RSA rates using the unfiltered supernatant and the filtered fractions as shown in Figure 4. This indicates that the signalling molecules were able to pass through the filter and were not retained. Untargeted metabolomics analysis of the supernatant could provide further insights to this question. We have added a sentence

in the main text (line 381) to emphasize this point: 'An untargeted metabolomics analysis should be performed to unravel the nature of the stress signal detected in our experiments.'

Otherwise, it is a nice study, and it is well presented.

Reply: Many thanks.

Reviewer #2 (Comments for the Author):

The study by Paton et al. presents an interesting new insight on ART-Pf interactions with potential clinical significance. The results clearly indicate that the parasite's response to ART can be modified by exposure to conditioned culture medium prepared by killing parasites with CQ and collecting the supernatant ('death signal'). The authors suggest the death signal released by dying trophozoites is a cell-cell signaling mechanism that leads to enhanced ring-stage survival after ART exposure independent of pfc13 genotype. The possibility that this signal is mediated EVs was eliminated size-exclusion spin columns and the retained potency of different size-exclusion fractions was interpreted by the authors to indicate the death signal has multiple chemical mediators of different sizes. These have potential important implications on evolving multigenic ART-R mechanisms as such a feedback loop independent of K13 could be a driver for selection of other types of parasite tolerance to ART exposure. This interpretation is consistent with other recent studies that have demonstrated a clear link between induction of parasite stress responses and altered sensitivity to ART. Overall this is a superb study that adds to our understanding of factors influencing parasite responses to antimalarial drugs.

Reply: We are very grateful to reviewer #2 for this positive and thoughtful assessment. Our data provided evidence that the increase in RSA rates is likely caused by the ring stage growth arrest induced by the stress signal. We did not specifically show that the stress signal is composed of multiple compounds but suggested this as a possibility. To this end, we have changed the sentence (lines 116-118) to read as follows: 'We also provide evidence that the signal is mediated by potentially multiple low molecular weight (<3 kDa) molecules, which would rule out a role for extracellular vesicles (EVs).' The sentence (lines 326-328) has also been revised: 'According to our results (Figure 4), we have identified such extracellular signals as a potential group of molecules <3 kDa.'

Major concern:

What is not clear from the study is the origin and nature of the active ingredient(s) of the death signal. The description of how the supernatant was prepared is inadequate. It is essential to include a detailed protocol in the Methods or as a separate supplement, so others can repeat the conditions used. This would include catalog numbers, etc... Include whether the supernatant was filtered or centrifuged to remove membrane fragments.

Reply: We agree with the comment of Reviewer #2 regarding the unknown nature of the stress signal. The focus of our work in this study was mainly to demonstrate the existence of such a signal *in vitro*. The protocol used to prepare the supernatant is described in detail in the supplementary information (1. Stress-induced medium mediates delayed growth of *P. falciparum* ring-stages *in vitro*). We agree that this information should be available in the main text (Materials and methods section) for

clarity and convenience. This information has now been added in the main text (line 430) in the paragraph: "Stress induced and supernatant prepared M_NF54_CQ20h". *P. falciparum* strain NF54 (chloroquine-sensitive) was grown and maintained asynchronously as described previously. When 1% parasitaemia was reached, infected erythrocytes were treated with chloroquine (Sigma) at a final concentration of 200 nM for 20 h and the culture was incubated at 37°C, 90% N₂, 5% CO₂, 5% O₂. The culture was then centrifuged (2000 rpm, 5 min, room temperature) and the culture medium (i.e. the supernatant) was collected in a 50 mL Falcon tube and stored at 37°C in a water bath until use. The supernatant was not filtered, in order to retain all components and molecules from the chloroquine-induced stress medium culture. Supernatants were prepared extemporaneously for each experiment. Chloroquine-induced stress in the NF54 parasite line was assessed by Giemsa-stained blood smears before and after chloroquine exposure (see Supplementary Information - 1. Stress-induced medium mediates delayed growth of *P. falciparum* ring-stages *in vitro*).

Minor concerns:

1. It would be useful to include counts on gametocyte abundance in addition to asexual stages.

Reply: No gametocytes were observed in the cultures. We cultured early ring stages (0-3 hpi) for 72h, making it impossible to observe gametocytes.

2. Provide the rationale how was it decided on the stress-induced medium conditions used in this study.

Reply: We chose chloroquine as the stress inducer for several reasons: First, the use of this drug makes the induced stress easily reproducible; second, we used chloroquine because this drug acts on mature stages by inhibiting haem crystallization - so the NF54 asynchronous parasite population was gradually killed as the parasites developed into mature stages during the 20 hour culture; thirdly, the parasite strains exposed to the supernatant were resistant to chloroquine (see Table 1 in the Supplementary Data), ruling out any possible effect of chloroquine on these strains when exposed to M_NF54_CQ20h supernatant.

3. What was RBC phenotype of blood used for culture, and sera?

Reply: We used O+ blood collected from healthy human donors at the Strasbourg blood bank (Etablissement Français du Sang). The AB serum used was also obtained from the Strasbourg blood bank

4. Since the effector molecules were not identified, the results from the size exclusion analysis could also be explained by a single small molecule present as monomers and polymers or aggregated. Similarly, the candidate molecule may be a larger molecule that degrades while the smaller products still retain activity. Such alternatives should be included in the discussion. Also, the authors cannot exclude the possibility that the active molecules are released from modified RBC of the of parasite-infected cells.

Reply: We thank the reviewer for these comments. We fully agree with the two first sentence and we have added this suggestion in the discussion (see page 11, lines 350-354). Relating to the last sentence, the data presented in Figures 2 and 5 provide

evidence that the RBC is not involved in the release of the stress signal.

5. L317 - "secrete" implies a specific biological process and there are data provided to support this is the active mechanism involved. "released" would be a more accurate term to use until the process is better defined. The accompanying speculation on similarities to other organisms should be moved out of the Results and into the Discussion.

Reply: As suggested, the term "secrete" has been replaced by "release" (lines 119 and 314). The words (line 318) '[...] as has been reported in bacteria and yeast' have been removed.

6. The data presented do not adequately support interpretation of a dose-dependent response, and the authors should stop at "threshold dependent".

Reply: We have changed the term "dose-dependent" to "threshold-dependent" (line 323). We also add "in a threshold-dependent manner" in the final sentence of the paper (line 384) to clarify our results. In addition, the threshold-dependent response with the supernatant is mentioned in lines 221-222 and 323.

7. Instead of reference #36 (a review), primary references should be cited for these studies.

Reply: Reference #36, used to describe the response of *P. falciparum* to thermal stress, is now accompanied by two other references (Ref. #28 and 29) cited in the discussion (lines 3336-337).

8. The end paragraphs are highly speculative considering no molecules mediating the altered responses were identified, no mechanism was identified, nor were any critical metabolic pathways defined. These paragraphs should be deleted or significantly revised without the speculation.

Reply: As requested by Reviewer #2, we have revised the last few paragraphs of the Discussion.

- Line 358, we have replaced 'explanation' by 'hypothesis'.
- Line 375-377, we have reworded the sentence which reads as: 'However, the signals may interact with a surface receptor, as such a mechanism has already been described (46) or pass through the erythrocyte membrane in an active (transporter protein) or passive manner.'
- Line 381, we have added the following sentence: 'An untargeted metabolomics analysis should be performed to unravel the nature of the stress signal detected in our experiments.'

While we agree with reviewer #2 that the stress signal molecule was not identified in this study, we demonstrate its existence in vitro. We believe that the data presented herein support the hypothesis mentioned in the discussion, as we know that the stress signal is produced by dying mature stages and has a molecular weight < 3 kDa. We have excluded the involvement of EVs (Figure 4), which suggests that the stress inducer is a small molecule(s).

Otherwise, it is a nice study, and it is well presented.

Reply: Many thanks.

Re: Spectrum03500-23R1 (Drug-induced stress mediates Plasmodium falciparum ring-stage growth arrest and reduces in vitro parasite susceptibility to artemisinin)

Dear Dr. Didier Menard:

Your manuscript has been accepted, and I am forwarding it to the ASM production staff for publication. Your paper will first be checked to make sure all elements meet the technical requirements. ASM staff will contact you if anything needs to be revised before copyediting and production can begin. Otherwise, you will be notified when your proofs are ready to be viewed.

Sincerely,
Anat Florentin
Editor
Microbiology Spectrum

Reviewer #1 (Comments for the Author):

I am happy with the revised manuscript. The concerns raised in the previous submission have been addressed in the revised manuscript.

Reviewer #2 (Comments for the Author):

The responses to the critique are acceptable and no additional changes are recommended. It is an excellent study.